# Preparation of a Heterogeneous Catalyst CuO-Fe₂O₃/CTS-ATP and Degradation of Methylene Blue and Ciprofloxacin

**Ting Zhang \***, **Wenhui Li, Qiyang Guo, Yi Wang** and **Chunlei Li**

School of Petrochemical Engineering, Lanzhou University of Technology, 287 Langongping Road, Lanzhou 730050, China; lwh18298324665@163.com (W.L.); guoqiyang@gmail.com (Q.G.); wangyi@lut.edu.cn (Y.W.); licl@lut.edu.cn (C.L.)
\* Correspondence: zhangting@lut.edu.cn

**Abstract:** A heterogeneous particle catalyst (CuO-Fe₂O₃/CTS-ATP) was synthesized via injection molding and ultrasonic immersion method, which is fast and effective. The particle catalyst applied attapulgite (ATP) wrapped by chitosan (CTS) as support, which was loaded dual metal oxides CuO and Fe₂O₃ as active components. After a series of characterizations of catalysts, it was found that CuO and Fe₂O₃ were successfully and evenly loaded on the surface of the CTS-ATP support. The catalyst was used to degrade methylene blue (MB) and ciprofloxacin (CIP), and the experimental results showed that the degradation ratios of MB and CIP can reach 99.29% and 86.2%, respectively, in the optimal conditions. The degradation mechanism of as-prepared catalyst was analyzed according to its synthesis process and ·OH production, and the double-cycle catalytic mechanism was proposed. The intermediate products of MB and CIP degradation were also identified by HPLC-MS, and the possible degradation pathways were put forward.

**Keywords:** attapulgite; chitosan; metal oxides; heterogeneous catalyst; mechanism; pathway





## 1. Introduction

Refractory organics (ROS), such as dyes, antibiotics, pesticides, etc., always threaten the survival and health of animals and human beings [1–4]. Methylene blue (MB) is a common dye in the printing and dyeing industry, and it is also used as a chemical indicator and biological dye [5]. Ciprofloxacin (CIP), as one of the third generation of antibacterial medicines, is widely used for fighting against various organ infections due to its broad-spectrum antibacterial activity. Due to a high percentage (60%–90%) of CIP discharge into the natural environment, nowadays CIP is considered to be one of the most important emerging pollutants due to their inhibition of bacterial activities [6–10]. Ciprofloxacin is generally difficult to biodegrade and enters the environment in the form of prototypes or metabolites [11,12], which poses a serious threat to the health of humans and aquatic organisms.

As the most promising methods, advanced oxidation processes, such as Fenton reactions, photocatalysis, electrocatalysis, ozone oxidation, etc., are often used to deal with refractory organic compounds in water which can produce a lot of hydroxyl radicals with strong oxidizing activity [13–18]. Fenton reactions, including all kinds of Fenton-like reactions, have become a research hotspot in the field of water treatment due to its high degradation efficiency, mild reaction conditions, low energy consumption, and simple operation [19–22]. In particular, the heterogeneous Fenton method has the advantages of wide pH range, low H₂O₂ and catalysts consumption, easy separation, and no secondary pollution with high removal ratios [23–26].

Attapulgite clay is a crystalline hydrated magnesium aluminium silicate mineral with unique layer-chain structure. It is made up of two layers of silicon-oxygen tetrahedrons that are sandwiched with a layer of magnesium (aluminium) oxygen octahedron. Attapulgite

has various excellent properties: good adsorption and catalytic properties because it is more porous, has a high specific surface area, and is rich in active groups [27–30]. Chitosan is the product of natural polysaccharide chitin by removing some acetyl groups, and it has many physiological functions, such as biodegradability, biocompatibility, anti-bacteria, anti-cancer, lipid-lowering, immunity enhancement, etc. [31–33]. Chitosan molecules have alkaline aminopolysaccharides with positive charge, which make them have good adsorption properties, and chitosan is widely used in textile, environmental protection, agriculture and other fields as a sorbent [31–33]. Introducing chitosan into attapulgite can effectively prevent the agglomeration of attapulgite, and attapulgite-chitosan composites are often used to treat wastewater [34–38]. For examples, Liang [35] et al. coated magnetic $Fe_3O_4$/APT nanoparticles modified by aminopropyltrimethoxy silane (APTS) with chitosan gel beads, and found that the adsorption capacity of this catalyst was much higher than that of attapulgite alone. Pan et al. [36] prepared low-cost chitosan/attapulgite composites by the self-assembly method for the removal of uranium in aqueous solution. Similarly, Liao et al. [37] prepared a novel three-dimensional porous polydopamine functionalized attapulgite/chitosan (AT@PDA/CS) aerogel for the adsorption of uranium, and when the pH of the solution was 5, it reached adsorption equilibrium in 40 min. Shi et al. [38] prepared divalent copper ion imprinted chitosan/attapulgite polymer, which showed good selectivity to copper ions, and the maximum experimental adsorption value reached 35.20 mg/g.

Most of the recent research about chitosan/attapulgite composites focused on adsorption materials, and few of those concentrated on catalytic materials. In this study, in order to study the catalytic abilities of attapulgite/chitosan composites, a catalyst CuO-$Fe_2O_3$/CTS-ATP was prepared by combining attapulgite with chitosan as a carrier and loading metal oxides on CTS-ATP support with an ultrasonic impregnation method. Dyes and antibiotics are refractory organics with complex and stable structures, for investigating the catalytic performance on ROS of the prepared catalyst, methylene blue (MB) from dyes and ciprofloxacin (CIP) from antibiotics were selected as target pollutants for their hard-to-degrade properties and harm to humans and the environment [39,40], and the catalytic abilities of prepared catalyst were evaluated when using it to degrade MB and CIP.

## 2. Materials and Methods

### 2.1. Materials

Methylene blue ($C_{16}H_{18}ClN_3S$), ciprofloxacin ($C_{17}H_{18}FN_3O_3$), chitosan, hydrochloric acid, concentrated sulfuric acid (98%), hydrogen peroxide (30% $w/w$), anhydrous ethanol, sodium hydroxide, $Fe(NO_3)_3 \cdot 9H_2O$, and $Cu(NO_3)_2 \cdot 3H_2O$ were all supplied by manufactures and analytically pure. Attapulgite clay (92%–98%) was obtained from Jiangsu province in China. The water used in experiments is all deionized water.

### 2.2. Preparation of CuO-$Fe_2O_3$/CTS-ATP

Catalyst CuO-$Fe_2O_3$/CTS-ATP was prepared by combining attapulgite with chitosan as a carrier and loading metal oxides on CTS-ATP support with the ultrasonic impregnation method. Firstly, 1 mL acetic acid was added into 99 mL deionized water to prepare 1% wt acetic acid solution, and 2 g attapulgite clay and 2 g chitosan were added into the prepared acetic acid solution, with magnetic stirring at room temperature until the two were fully mixed. Then, the mixed CTS-ATP gel solution was evenly poured into a 50 mL syringe and was dripped into 200 mL NaOH (1 mol/L) solution dropwise, and small CTS-ATP gel particles were dispersed in the solution. Then, 5 mL of 2.5% wt glutaraldehyde solution were added into the above solution. After an hour's reaction, the prepared CTS-ATP gel particles were filtered and put into the drying oven at 100 °C for an hour, then put into the tube furnace and roasted with nitrogen atmosphere (15 mL/min) at 400 °C for 2 h, so black particles of CTS-ATP carrier with a diameter of 1 mm were obtained. Add the particles into a certain concentration of ferric nitrate and copper nitrate (1:1) mixing solutions (5% wt for MB degradation and 15% for CIP degradation, Supplementary Materials Figure S1) and

impregnated with ultrasonic for 50 min (Figure S2). Finally, the particles were taken out and rinsed with deionized water, and roasted for 1 h (temperature 200 °C, nitrogen inlet rate 15 mL/min) in the tube furnace to obtain heterogeneous catalyst $CuO-Fe_2O_3/CTS-ATP$.

### 2.3. Characterization

The crystalline structure and stability of ATP, $Fe_2O_3/CTS-ATP$, and $CuO-Fe_2O_3/CTS-ATP$ were identified by the Palytical X 'PERT PRO X-ray diffraction (XRD, Almelo City?, The Netherlands) analyzer, operating voltage and current were 40 kV and 150 mA, the diffraction Angle was 5°–80°, and the scanning step was 0.02°/s. All samples above were tested by a Nicoletavtar 360 FT-IR spectrometer (Almelo, Madison, WI, USA), and the bonding mode between the compounds and elements was analyzed. The scanning wave range is 4000–400 $cm^{-1}$.The surface morphology of samples were observed by a JEOL JSM-6701F scanning electron microscopy (SEM, JEOL, Tokyo, Japan). The acceleration voltage was 20 kV, and the samples' surface needs Pt sprayed for good electrical conductivity. An energy dispersive spectrometer (EDS QUANTAX) (produced by Bruker Corporation, Rheinstetten, Germany) was used to analyze the content and element mapping of the main elements in the samples. The $N_2$ adsorption and desorption isotherms of the samples were measured at −197 °C by the ASAP2010 specific surface area tester manufactured by Micromeritics, Norcross, GA, USA. The specific surface area, pore volume, and pore size were calculated by the BET equation. The qualitative and quantitative analysis of elements on the surface of materials, as well as the analysis of chemical valence states and valence electron states, were tested by the PHI5702 X-Ray photoelectron spectrometer (XPS) (produced by the American Physical Electronics Company, Chanhassen, MN, USA). The X-ray emission sources are Mg, Al double-anode target, and Al monochromator target.

### 2.4. Degradation Experiments of MB and CIP by the Catalysts

In addition, 1 g/L methylene blue and 250 mg/L ciprofloxacin stock solutions were prepared for later use. An appropriate amount of MB or CIP stock solutions and a certain amount of deionized water were added into a 250 mL flask to obtain MB or CIP solutions with different concentrations. Add catalyst and 30% $w/w$ hydrogen peroxide into the flask, stirring it at a certain temperature. Take water samples every 10 min, and measure the absorbance of the solution by UV-1900 UV-visible spectrophotometer (Aoyi Instrument Co., Ltd., Shanghai, China). In order to reduce experimental error, one blank and three same samples were tested at the same conditions in each experiment group. The removal ratios of MB or CIP solutions under different conditions can be calculated by Equation (1):

$$\text{Removal ratio } (\%) = (C_0 - C_t)/C_0 \times 100\% \tag{1}$$

$C_0$ is the initial concentration of MB or CIP solutions before treatment, and $C_t$ is the concentration at reaction time t.

### 2.5. OH Measurement and Intermediate Product Determination

·OH can react with salicylic acid and produce a chemical substance called 2, 3-dihydroxybenzoic acid with maximum absorption peak at the wavelength of 510 nm. Thus, the ·OH concentration produced by $CuO-Fe_2O_3/CTS-ATP/H_2O_2$ system can be measured by a UV1900 type ultraviolet-visible spectrophotometer with ·OH spectrophotometry at λ of 510 nm.

The intermediate products of MB and CIP degraded by $CuO-Fe_2O_3/CTS-ATP/H_2O_2$ system were determined by high performance liquid chromatography-mass spectrometry (HPLC-MS). Chromatography and mass spectrometry were performed by Agilent 1200 high performance liquid chromatography (Agilent, Santa Clara, CA, USA) and a 6130 quadruple mass spectrometer (Agilent, Santa Clara, CA, USA), respectively. The separation column is ZORBAX Eclipse Plus C18 (Analytical 4.6 mm × 150 mm 5-Micron). The operating conditions for the analysis of MB and CIP intermediate products were as follows: the mobile phase A was 0.2% formic acid high pure water, mobile phase B was acetonitrile, the

temperature of separation column was 30 °C, the injection volume was 10 µL, the Electron Spray Ionization (ESI) in negative ion mode was adopted as the mass spectrometer detector, the capillary current was 5 nA, the flow rate of dry gas was 10 L/min, and the dry gas temperature was 350 °C.

## 3. Results and Discussion

### 3.1. Characterization of Samples

To investigate the crystalline structure and stability of ATP, $Fe_2O_3$/CTS-ATP and CuO-$Fe_2O_3$/CTS-ATP composites and powder X-ray diffraction patterns were recorded by an X-ray diffractometer. It can be seen from Figure 1a that all samples showed diffraction peaks at 2θ of 8.5°, 19.5°, 27.8°, 34.7°, and 35.4°, corresponding to the diffraction peaks of attapulgite crystal structure, indicating that ATP's crystal structure was not damaged when CTS and metal oxides were introduced into ATP while the disorder degree of ATP crystal increased due to the changes of basal peaks. The diffraction peaks at 2θ of 24.2°, 33.2°, 40.9°, 49.6°, and 54.2° belong to $Fe_2O_3$ [41], and the diffraction peaks of 35.6°, 38.7°, 49.1°, 53.5°, 58.3°, and 65.9° are characteristic peaks of CuO [41,42]. These indicate that the metal oxides are successfully loaded on the surface of CTS-ATP. The infrared spectrums of the samples were shown in Figure 1b. The strong absorption peak near 3430 cm$^{-1}$ is a O–H bending vibration peak of attapulgite. The peak near 1630 cm$^{-1}$ is H–O–H stretching vibration peak of water. The peak around 1482 cm$^{-1}$ is C=O stretching vibration peak. The bending vibration peak of Si–O appears at 1040 cm$^{-1}$. The wave number peaks at 740 cm$^{-1}$ and 476 cm$^{-1}$ are stretching vibration peaks of metal oxides Cu–O bond and Fe–O bond on the surface of $Fe_2O_3$/CTS-ATP and CuO-$Fe_2O_3$/CTS-ATP composites, indicating that both iron and copper are successfully loaded on the surface of carrier in the form of oxides. The peaks of Fe-O and Cu-O may not be too obvious due to low loading percentage.

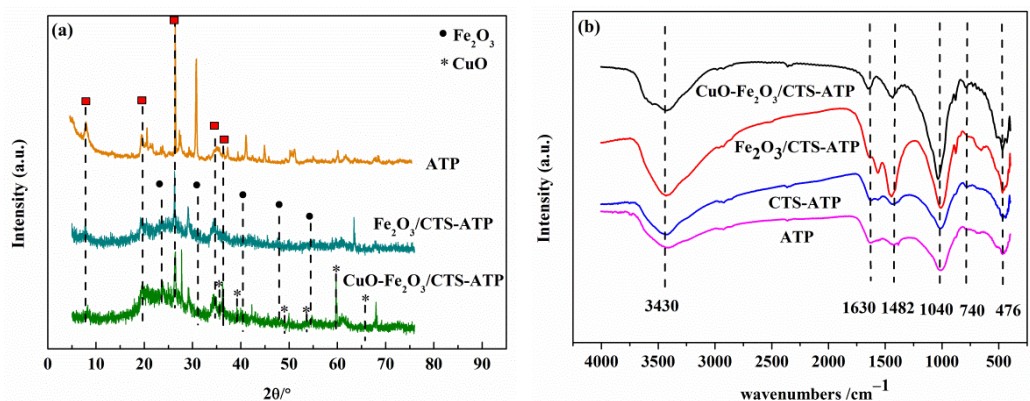

**Figure 1.** XRD (**a**) and FT-IR (**b**) patterns of the samples.

To compare the structure and morphology characteristics of ATP, CTS-ATP, $Fe_2O_3$/CTS-ATP, and CuO-$Fe_2O_3$/CTS-ATP, scanning electron microscopy (SEM), Mapping, and EDS were used to observe and analyze the element content and distribution on the sample surface. The results are shown in Figure 2. Figure 2a–d are SEM images of the samples, which showed that ATP has rod crystal structure (a), and ATP was wrapped by CTS when ATP mixed with CTS (b), and further loading $Fe_2O_3$ or $Fe_2O_3$/CuO and calcined to form active sites and holes (c,d). The purpose of introducing CTS into ATP is to regulate the surface morphology of the catalyst and make the loaded active materials more uniform on support, which can be proved by the Mapping results of Figure 2e,f, showing that the loaded Fe or Fe/Cu on the surfaces of the catalysts are relatively uniform. Figure 2g–I are DES results of the samples, and, from Figure 2h,I, it is also clearly shown that Fe or Fe/Cu were successfully loaded on the catalysts.

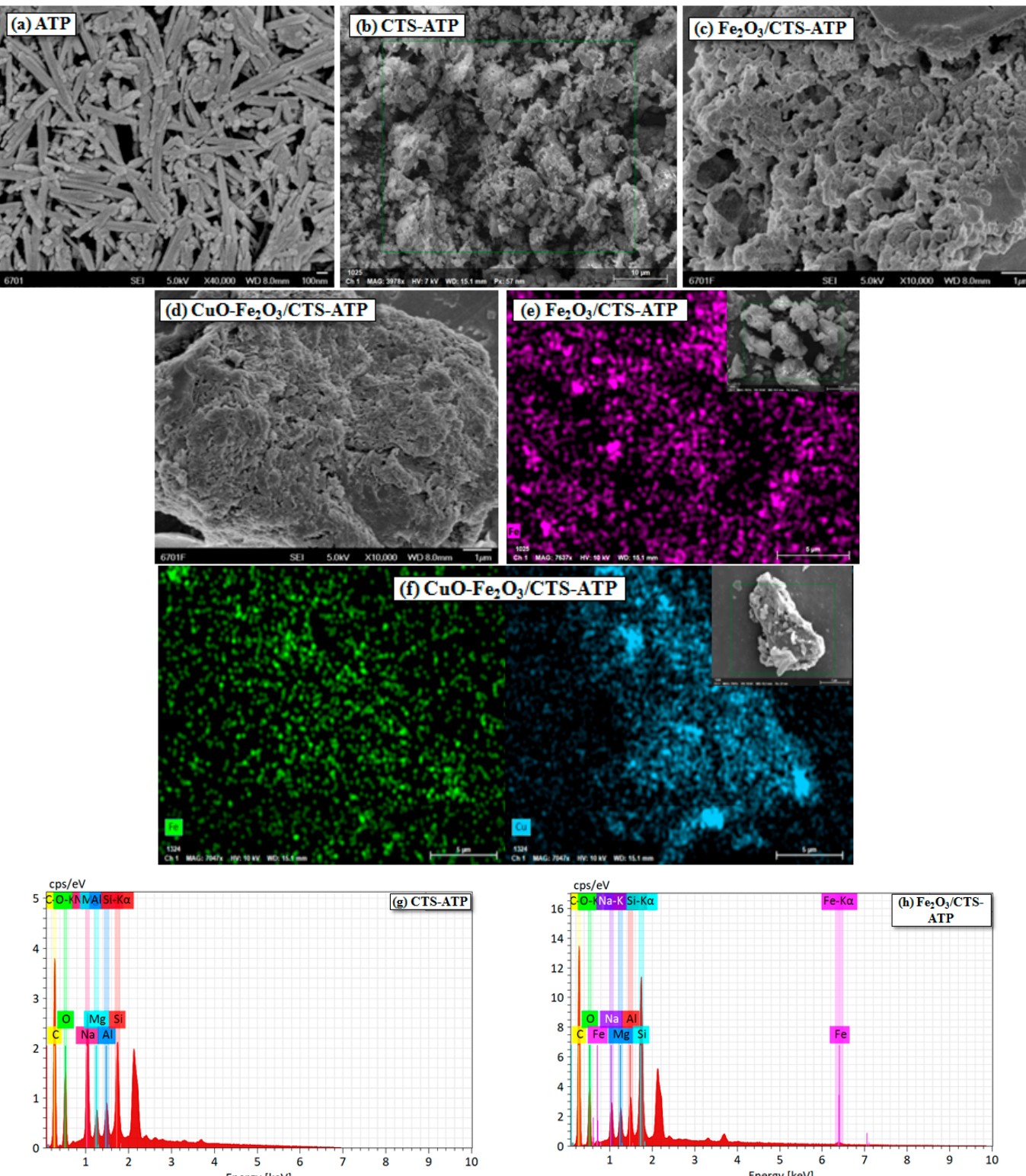

**Figure 2.** *Cont.*

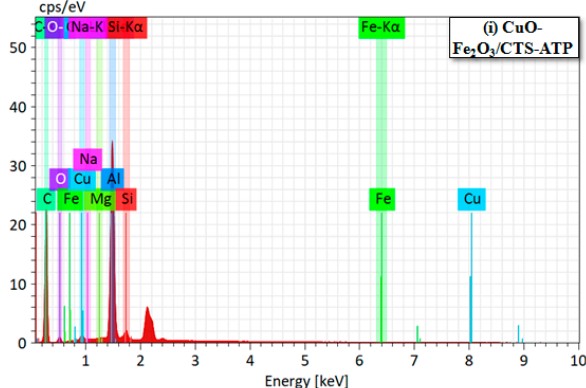

**Figure 2.** SEM (**a–d**); Mapping (**e,f**); and EDS (**g–i**) images of the samples.

Table 1 shows the specific surface area, pore volume, and pore size of ATP, $Fe_2O_3$/CTS-ATP and CuO-$Fe_2O_3$/CTS-ATP. The BET specific surface area and pore volume of $Fe_2O_3$/CTS-ATP (21.977 m$^2$/g and 0.061 cm$^3$/g) and CuO-$Fe_2O_3$/CTS-ATP (23.23 m$^2$/g and 0.095 cm$^3$/g) decreased dramatically compared to those of ATP (93.620 m$^2$/g and 21.509 cm$^3$/g) due to the wrapping of CTS on ATP, while the pore size of $Fe_2O_3$/CTS-ATP (11.013 nm) and CuO-$Fe_2O_3$/CTS-ATP (14.271 nm) increased greatly compared to that of ATP (0.227 nm), which meant that mesopores are formed on the surface of the catalyst during the roasting process [43].

**Table 1.** BET measurement results of samples.

| Samples | BET Surface Area/m$^2$·g$^{-1}$ | Pore Volume/cm$^3$·g$^{-1}$ | Pore Size/nm |
|---|---|---|---|
| ATP | 93.620 | 21.509 | 0.227 |
| $Fe_2O_3$/CTS-ATP | 21.997 | 0.061 | 11.013 |
| CuO-$Fe_2O_3$/CTS-ATP | 23.23 | 0.095 | 14.271 |

Figure 3 is nitrogen adsorption and desorption (A-D) curves as well as pore size distribution diagrams of $Fe_2O_3$/CTS-ATP and CuO-$Fe_2O_3$/CTS-ATP. The A-D curves of the two samples belong to type III isotherms and H3 magnetic loop according to IUPAC classification, representing their mesoporous structure, which is in accordance with the characteristic result of SEM images. The attached pore size distribution diagrams showed that the two samples had the same pore size distribution, while CuO-$Fe_2O_3$/CTS-ATP had more pores with the size ranging from 50–100 nm, and the pore structure belongs to the slit pore.

In order to analyze the chemical state of various elements of the samples, XPS tests of the catalysts were performed. The total spectra and characteristic spectra of the elements are shown in Figure 4. The survey spectra of the samples confirmed that Fe and Cu were present in these two samples besides C, O, and Si, while the peaks of Fe and Cu were very weak due to the low loading deposition level (Figure 4a). In Figure 4b, the peaks at 284.8, 286.2, and 286.6 eV of the samples were C1s bonded to H, N, and O, which might be formed during the roasting of the samples. In Figure 4c, the peaks at 531 eV of the samples were relevant to O1s bonded to Fe, Cu, and Si, whereas the peaks at higher binding energy of 532.2 eV were ascribed to the presence of OH$^-$ group [44,45] on the catalysts' surface. High resolution XPS spectrum of Fe2p in Figure 4d located at the binding energies of 713.2 (713.6) eV and 726.8 (726.4) eV were indicative of Fe2p3/2 and Fe2p1/2 of the oxidation state $Fe^{3+}$ in samples [46]. The characteristic peaks of Cu2p3/2 and Cu2p1/2 at 933.1 and 952.4 in the spectrum of Cu2p in Figure 4e, accompanied by a series of satellite vibration peaks, indicated the presence of $Cu^{2+}$ on the surface of the catalyst, which also agreed with the mapping results.

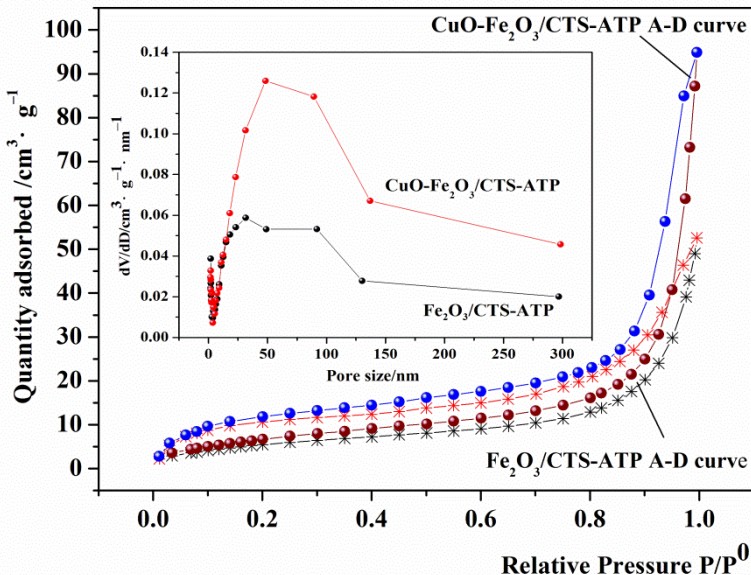

**Figure 3.** Nitrogen adsorption–desorption curves and pore size distribution diagrams of Fe$_2$O$_3$/CTS-ATP and CuO-Fe$_2$O$_3$/CTS-ATP.

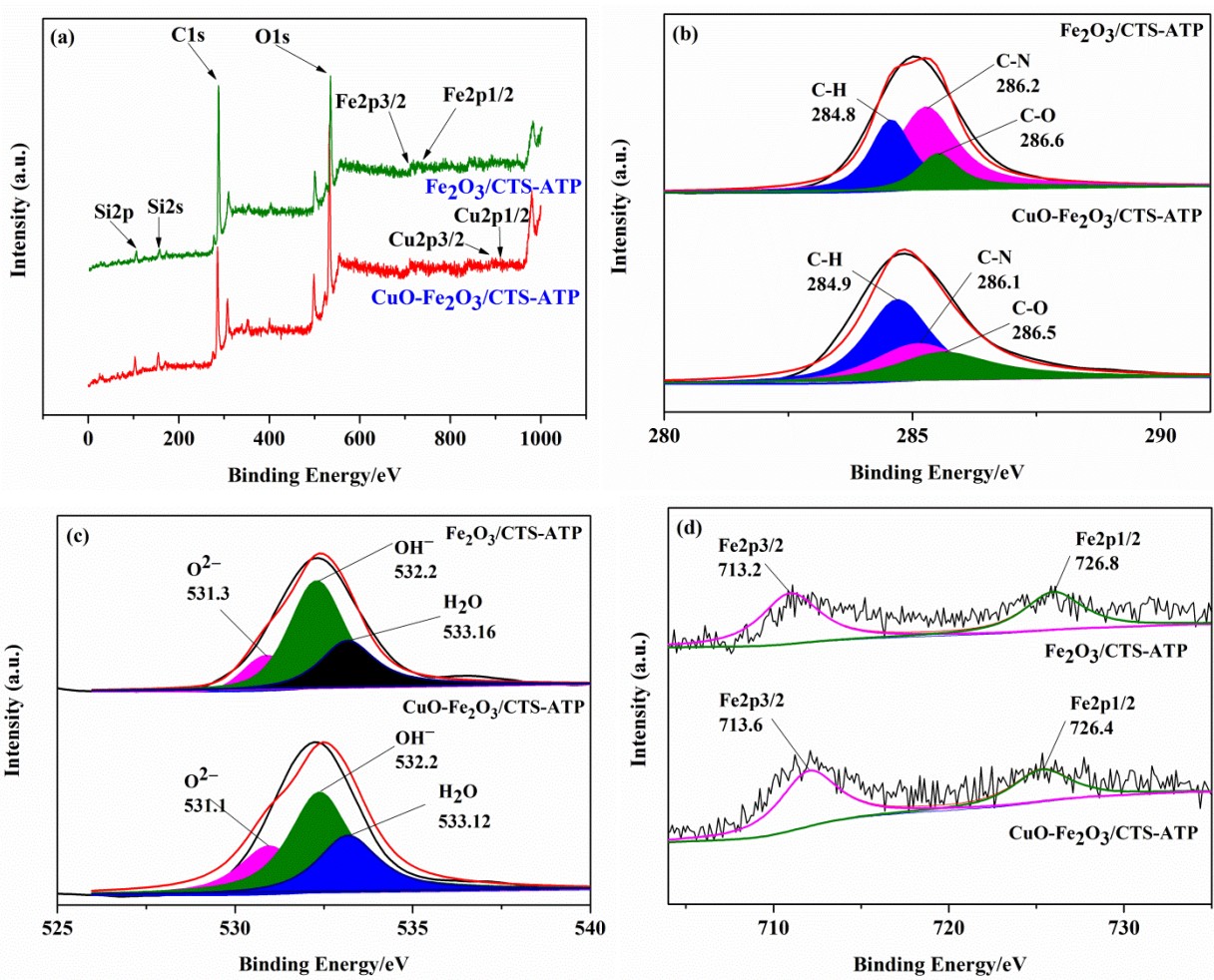

**Figure 4.** *Cont.*

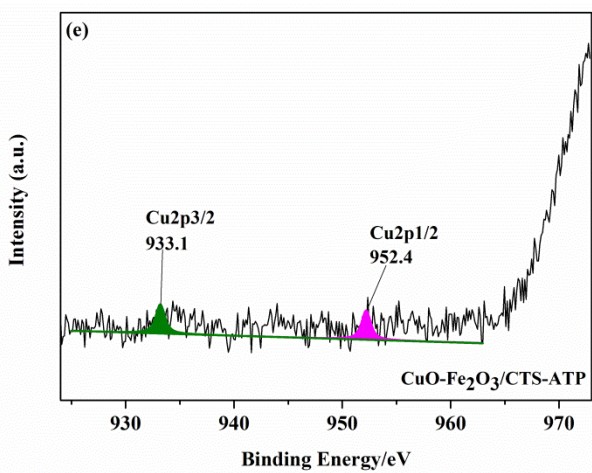

**Figure 4.** XPS full spectra (**a**), C1s spectra (**b**), O2p spectra (**c**), Fe2p spectra (**d**), and Cu2p spectra (**e**) of prepared catalysts.

*3.2. MB and CIP Degradation by Prepared Catalysts*

Figure 5 shows the effects on the MB and CIP removal ratios in five different systems (only $H_2O_2$ existing, only $Fe_2O_3$/CTS-ATP or CuO-$Fe_2O_3$/CTS-ATP existing, $Fe_2O_3$/CTS-ATP/$H_2O_2$ system or CuO-$Fe_2O_3$/CTS-ATP/$H_2O_2$ system), respectively. From Figure 5a,b, it can be seen that MB can obtain higher removal ratios than CIP at the same conditions. This means that oxidation of MB is easier than that of CIP, and the structure of CIP is more stable than that of MB. When only hydrogen peroxide exists, MB and CIP can obtain about 40% removal ratios after 60 min reaction because $H_2O_2$ has a certain oxidation ability but not too much (Figure 5a,b). When only $Fe_2O_3$/CTS-ATP or CuO-$Fe_2O_3$/CTS-ATP exists in the system, MB and CIP removal ratios were 60% or 50% separately due to the adsorption of catalysts and $O_2$ oxidation (Figure 5a,b). In the $Fe_2O_3$/CTS-ATP/$H_2O_2$ system, MB and CIP can obtain over 95% and 80% removal ratios after 60 min reaction because $Fe_2O_3$ catalyzed $H_2O_2$ to decompose ·OH, which had higher oxidation ability than $O_2$ and $H_2O_2$, whether in acidic or alkaline conditions, as can be seen in Figure 5c. In the CuO-$Fe_2O_3$/CTS-ATP/$H_2O_2$ system, MB and CIP can obtain over 99% and 90% removal ratios after 60 min reaction, which were higher and faster than those in the $Fe_2O_3$/CTS-ATP/$H_2O_2$ system, which also can be seen from the catalytic reaction rate constant k (Figure 5a,b). The above results can be explained by Figure 5d. In the $Fe_2O_3$/CTS-ATP/$H_2O_2$ catalytic system, Fe(III) and Fe(II) formed a cycle during the process of catalyzing $H_2O_2$ to generate ·OH, while, in the CuO-$Fe_2O_3$/CTS-ATP/$H_2O_2$ catalytic system, Fe(III) and Fe(II), and Cu(I) and Cu(II), formed a dual catalytic cycle to catalyze more $H_2O_2$ and generate more ·OH, which led to higher degradation ratios and faster degradation rates.

A series of single-factor (such as reaction temperatures, pH values, catalyst dosages, $H_2O_2$ concentrations, recycle times of the catalysts, et al.) experiments were performed to optimize the degradation conditions of a CuO-$Fe_2O_3$/CTS-ATP/$H_2O_2$ system (Figures S3). From 40 to 80 °C, as shown in Figure S3, MB, and CIP, can obtain high removal ratios, and, with the temperature increasing, their removal ratios were increasing due to the boosting of ·OH radicals at a higher temperature and a faster catalysis to decompose $H_2O_2$. As can be seen from Figure S4, MB and CIP obtained the highest removal ratios at pH values of 2 attributed to a high redox potential of ·OH in an acidic condition. In addition, they all reached over 60% removal ratios under all experimental pH values. Figures S5 and S6 indicated that degradation ratios of MB and CIP ascended with increasing catalyst dosage and $H_2O_2$ concentration, suggesting that the more catalysts and $H_2O_2$ there are, the more active sites OH radicals there are, resulting in high removal efficiencies. However, too many catalysts and $H_2O_2$ will lead to high costs, and too many $H_2O_2$ will react with ·OH radicals and release oxygen. Recycle degradation experiments of MB and CIP by CuO-$Fe_2O_3$/CTS-ATP catalysts were performed five times to test their durability (Figure S7).

The catalyst was only washed by distilled water after every run, and the reaction conditions maintained the same in every run. After five runs, MB and CIP also can obtain 98% and 86% degradation ratios, separately.

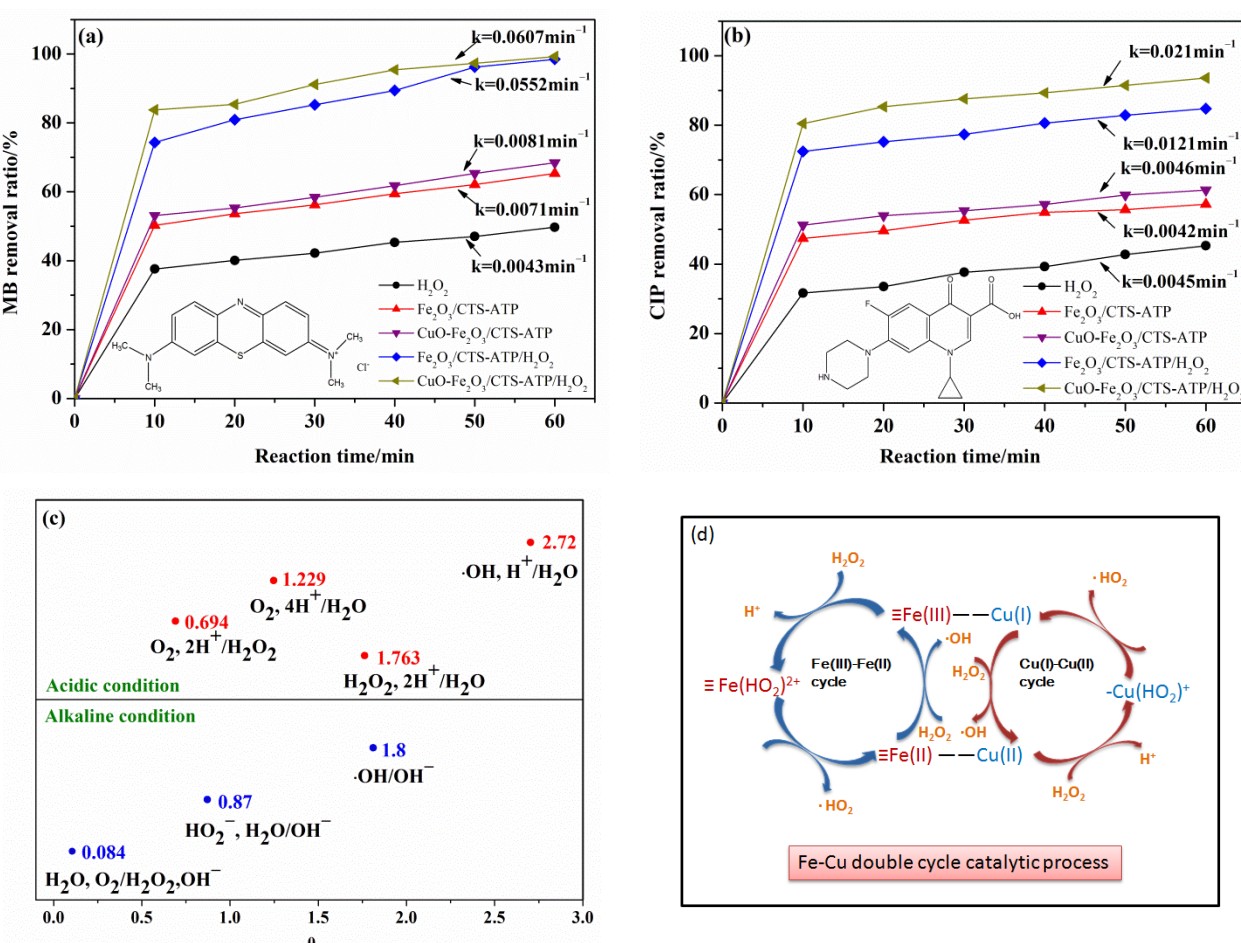

**Figure 5.** The effects on the MB and CIP removal ratios in different systems (**a**,**b**) and degradation mechanisms (**c**,**d**).

### 3.3. OH Concentrations Measurement

·OH concentrations in two catalytic systems were also detected, and the results were shown in Figure 6. It can be seen that the concentration of ·OH in CuO-Fe$_2$O$_3$/CTS-ATP/H$_2$O$_2$ system was more than that in the Fe$_2$O$_3$/CTS-ATP/H$_2$O$_2$ system, and the production rate of ·OH in former system was also higher than that in the latter. The results confirmed that these two systems could produce enough ·OH to degrade ROS, and CuO could be a good assistant catalytic component to enhance the ·OH production rate dramatically, which was coincident with the degradation mechanisms in Figure 5d.

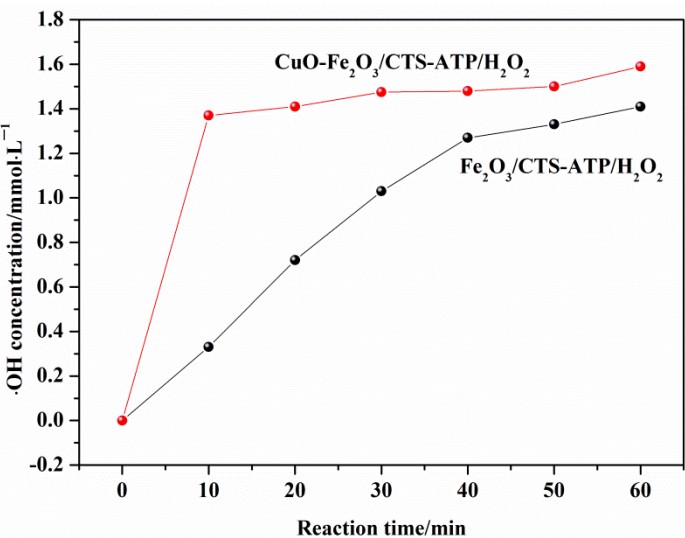

**Figure 6.** The trend of ·OH concentration in two systems with time.

### *3.4. The Pathways of MB and CIP Degradation by CuO-Fe$_2$O$_3$/CTS-ATP*

As can be seen in Scheme 1a, attapulgite and chitosan combined via hydrogen bond; thus, a core–shell structure with attapulgite as the core and chitosan as the shell was formed. After calcination, chitosan was carbonized and formed a net structure to wrap attapulgite. When loading active components, iron oxide and copper oxide first formed an alloy [47–49] and then combined with the carrier (CTS-ATP) in the form of a covalent bond, in order to obtain an efficient surface catalytic system.

In order to explore the intermediate products of MB and CIP in CuO-Fe$_2$O$_3$/CTS-ATP/H$_2$O$_2$, HPLC-MS was used to analyze the degradation products of MB and CIP at reaction times of 30 and 60 min, respectively. The treated-water samples were filtered and then sent into the separation column through the autosampler. When MB solution was catalyzed for 30 min, the signal peak of MB ($m/z$ = 284.19) at 14.6 min disappeared, and many small mass fragments appeared. The possible degradation path of MB molecule is shown in Scheme 1b, where the substance with $m/z$ = 242.82 is the product after MB demethylation under the attacking of ·OH [50]; then, the –S– bond and –N= were broken, and MB was broken into two pieces. After a series of degradations, the main identified products were benzoquinone (m/z = 107, RT = 3.161 min), hydroquinone ($m/z$ = 109, RT = 3.098 min), catechol ($m/z$ = 109, RT = 3.308 min), and resorcinol ($m/z$ = 109, RT 5.098 min).

The characteristic signal peak of the CIP was $m/z$ = 332.20 at 2.9 min, and it disappeared completely after the 30 min degradation of CIP. According to the identity results of CIP intermediate products by HPLC-MS, the possible degradation path is shown in Scheme 1c. Under the attacking of ·OH, CIP lost its carboxyl (–COOH) firstly, and the piperazine ring in CIP molecule underwent oxidation and ring opening; then, it was totally removed. After losing –C$_3$H$_5$N [51], another nitrogen-containing heterocycle of CIP was also opened. The oxidation process continued, and the carbon chain containing aldehyde and carboxylic acid on the benzene ring was broken to form fluorophenol. Finally, the benzene ring was opened.

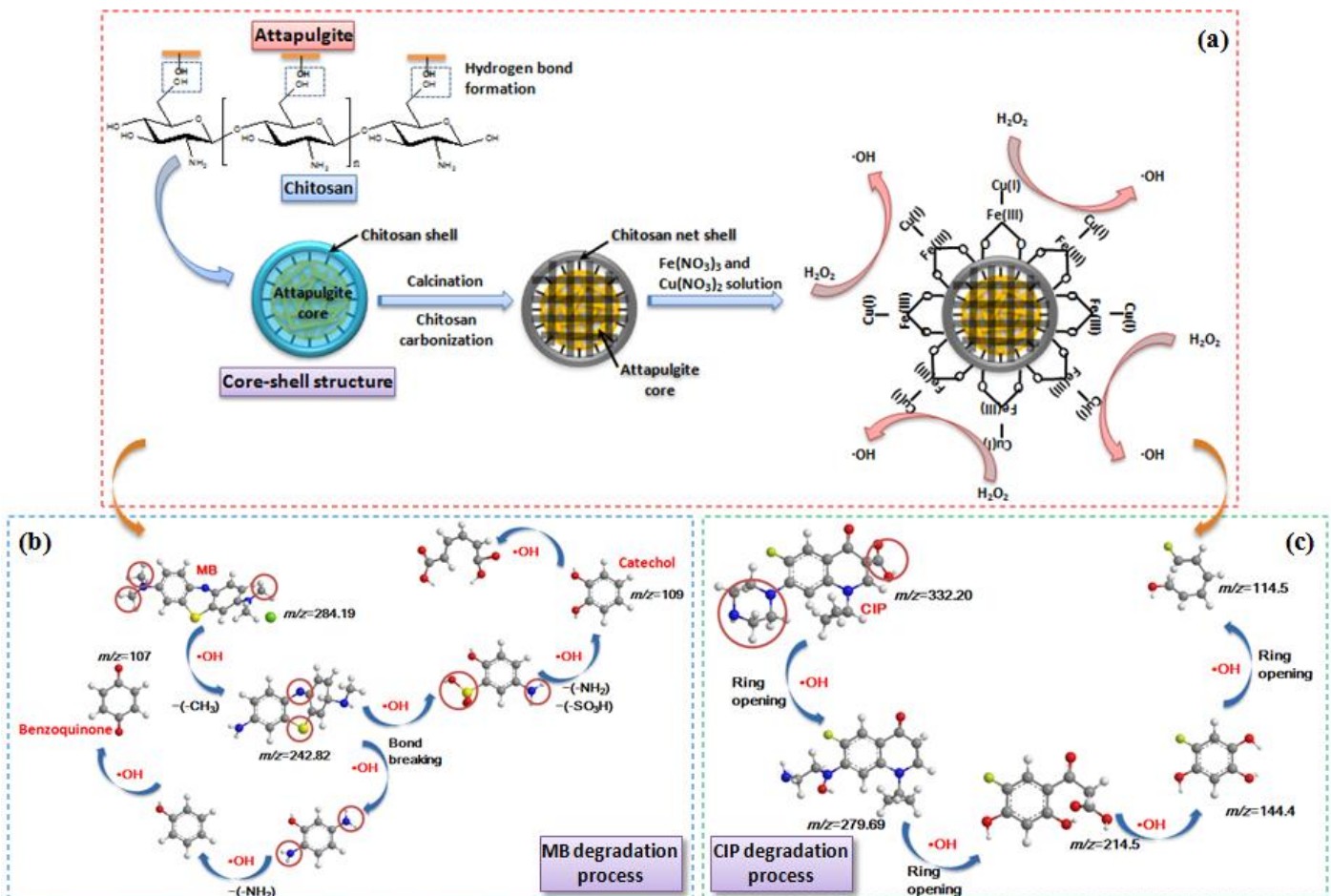

**Scheme 1.** Formation of catalytic system (**a**) and possible degradation pathways of MB (**b**) and CIP (**c**) molecules by the CuO-Fe$_2$O$_3$/CTS-ATP/H$_2$O$_2$ system.

## 4. Conclusions

In this study, a heterogeneous catalyst with dual support components (CTS-ATP) and dual active components (Fe$_2$O$_3$-CuO) was prepared for the degradation of MB and CIP. The catalysts were characterized by XRD, FI-IR, SEM, BET, and XPS, which showed that both Fe$_2$O$_3$ and CuO were successfully loaded on the surface of the carrier, and CTS as a functional material wrapped ATP would make the loaded active components more uniform, which greatly improves the catalytic degradation performance of CuO-Fe$_2$O$_3$/CTS-ATP. The preparation conditions and degradation conditions were optimized. Under the optimal preparation and reaction conditions, the degradation ratios of MB and CIP can reach 99.29% and 86.20%, respectively. The new degradation mechanisms and possible pathways of MB and CIP were proposed. The authors believed that Fe$_2$O$_3$ and CuO first formed an alloy and then combined with the carrier (CTS-ATP) in the form of covalent bonds, in order to obtain an efficient surface catalytic system. Fe(III) and Fe(II) and Cu(I) and Cu(II) formed a dual catalytic cycle to catalyze more H$_2$O$_2$ and generate more ·OH, which led to higher degradation ratios and faster degradation rates.

**Supplementary Materials:** The following supporting information can be downloaded at: https://www.mdpi.com/article/10.3390/coatings12050559/s1, Figure S1: Influences of precursor concentration (during catalyst preparation) on MB (a) and CIP (b) removal.; Figure S2: Influences of ultrasonic impregnation time (during catalyst preparation) on MB (a) and CIP (b) removal.; Figure S3: Influences of reaction temperature on MB (a) and CIP (b) removal.; Figure S4: Influences of pH values on MB (a) and CIP (b) removal.; Figure S5: Influences of catalyst dosages on MB (a) and CIP

(b) removal.; Figure S6: Influences of $H_2O_2$ concentrations on MB (a) and CIP (b) removal.; Figure S7: Influences of recycle times of prepared catalyst on MB (a) and CIP (b) removal.

**Author Contributions:** Conceptualization, T.Z. and Q.G.; Data curation, W.L. and Q.G.; Formal analysis, Q.G.; Funding acquisition, T.Z.; Investigation, W.L.; Methodology, T.Z. and Y.W.; Project administration, C.L.; Resources, Y.W. and C.L.; Supervision, C.L.; Validation, Y.W.; Writing—original draft, T.Z. and Q.G.; Writing—review & editing, T.Z. All authors have read and agreed to the published version of the manuscript.

**Funding:** This work was financially supported by Major Science and Technology Projects in Gansu Province of China (Grant No. 21ZD2JA001).

**Institutional Review Board Statement:** Not applicable.

**Informed Consent Statement:** Not applicable.

**Data Availability Statement:** Data is contained within the article or Supplementary Material.

**Conflicts of Interest:** The authors declare no conflict of interest.

**Abbreviations**

| | |
|---|---|
| ATP | attapulgite |
| CTS | chitosan |
| ROS | refractory organics |
| MB | methylene blue |
| CIP | ciprofloxacin |

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
