# Peer review of "Preparation of a Heterogeneous Catalyst CuO-Fe2O3/CTS-ATP and Degradation of Methylene Blue and Ciprofloxacin"

_coatings, doi:10.3390/coatings12050559_

Round 1

Reviewer 1 Report

Τhis work is based on the synthesis of CuO-Fe2O3/CTS-ATP and the degradation of methylene blue (MB) and ciprofloxacin (CIP).

This is an interesting work, nevertheless some revisions are needed in order to publish this manuscript.c

1 Actually, the authors cannot see the degradation of MB. What they see is it's decolorization. I kindly ask the authors to correct the term degradation to decolorization.

2 Could the authors calculate the kinetics constant k for each case? This is essential in order to compare their data with the literature.

3 Further discussion regarding adsorption-desorption curves is needed.

4 UV-Vis cannot detect degradation of CIP. UV-Vis can only detect decolorization. I kindly ask the authors to check comment #1. Correct the term degradation to decoration and calculate apparent rates k.

5 if the authors want to discuss about degradation they should characterize their stock solutions via FT-IR, Raman or other vibration techniques to detect chemical bonds.

6 some more references are needed both in the introduction and the analysis part of the manuscript.

7 the authors should provide the providers of chemicals and setups.

Author Response

Responses to reviewer 1’s comments:

Thanks for your useful comments and suggestions on the language and the structure of our manuscript. We have modified the manuscript accordingly, and the detailed corrections are listed below point by point:

1 Actually, the authors cannot see the degradation of MB. What they see is it's decolorization. I kindly ask the authors to correct the term degradation to decolorization.

--We have change “degradation” to “removal” because CIP is an antibiotic without color in solution, in order to illustrate the comparison of degradation effects between MB and CIP by as-prepared catalysts, we think the word “removal” is better.

2 Could the authors calculate the kinetics constant k for each case? This is essential in order to compare their data with the literature.

-- Yes, we have calculated the kinetics constant k for each case.

3 Further discussion regarding adsorption-desorption curves is needed.

-- We have discussed adsorption-desorption curves of as-prepared catalysts in page 5. We think that is enough for the as-prepared catalysts.

4 UV-Vis cannot detect degradation of CIP. UV-Vis can only detect decolorization. I kindly ask the authors to check comment #1. Correct the term degradation to decoration and calculate apparent rates k.

-- CIP is an antibiotic without color in solution, in order to illustrate the comparison of degradation effects between MB and CIP by as-prepared catalysts, we think the word “removal” is better.

5 if the authors want to discuss about degradation they should characterize their stock solutions via FT-IR, Raman or other vibration techniques to detect chemical bonds.

-- We employed LC-MS to determine of intermediate products (page 9) and discussed the degradation.

6 some more references are needed both in the introduction and the analysis part of the manuscript.

-- We have added some references for the introduction and analysis parts.

7 the authors should provide the providers of chemicals and setups.

-- We have provided the providers of chemicals and setups.

Reviewer 2 Report

A heterogeneous particle catalyst (CuO-Fe2O3/CTS-ATP) was synthesized chitosan: (CTS) and  attapulgite (ATP) were applied as support for metal oxides CuO and Fe2O3. The catalyst was characterized by Ft-IR, XRD and for absorption ability, and was used  to degrade methylene blue (MB) and ciprofloxacin (CIP) and the experimental conditions of the degradation were explored. The degradation mechanism  was also investigated. The intermediates of MB and CIP degradation were identified by HPLC-MS, and the possible degradation pathways were proposed. The results are sound, however, the ms is not too interesting.

The readers badly need the formula and structure of methylene blue and ciprofloxacin. Also a kind of „bridge” would be needed to justify the joint (in one paper) investigation of a dye and a drug.

Did the au-s performed blind probe experiments without the catalyst using only H2O2?

Au-s write: we „believe that Fe2O3 and CuO first formed alloy and then combined with the carrier (CTS-ATP) in the form of covalent bond, so as to obtain an efficient surface catalytic system”. Is there any proof for this assumption?

Au-s should consider to transfer a part of the illustrations ont he characterization of the catalyst into the Supplementary Info.

In overall, he ms may be accepted for the journal "Coatings" after the above minor revisions.

Author Response

Responses to reviewer 2’s comments:

Thanks for your useful comments and suggestions on the language and the structure of our manuscript. We have modified the manuscript accordingly, and the detailed corrections are listed below point by point:

  1. The readers badly need the formula and structure of methylene blue and ciprofloxacin. Also a kind of “bridge” would be needed to justify the joint (in one paper) investigation of a dye and a drug.

--We have added the formula and structure of methylene blue and ciprofloxacin in the manuscript. And we also gave a joint of investigation of a dye and a drug in our manuscript, which can be seen in page 2.

  1. Did the au-s performed blind probe experiments without the catalyst using only H2O2?

-- Yes, we did that, and the result was shown in Figure 5(a, b).

  1. Au-s write: we “believe that Fe2O3 and CuO first formed alloy and then combined with the carrier (CTS-ATP) in the form of covalent bond, so as to obtain an efficient surface catalytic system”. Is there any proof for this assumption?

--Several references can support my assumption, which can be seen as follows:

Xia Qin, Ziyuan Wang, Chengrui Guo, Rui Guo, Yue Lv, Mingran Li, Fulvic acid degradation in Fenton-like system with bimetallic magnetic carbon aerogel Cu-Fe@CS as catalyst: Response surface optimization, kinetic and mechanism, Journal of Environmental Management, 306 (2022) 114500.

Qixing Xia, Dongjie Zhang, Zhongping Yao, Zhaohua Jiang, Revealing the enhancing mechanisms of Fe–Cu bimetallic catalysts for the Fenton-like degradation of phenol, Chemosphere, 289 (2022) 133195.

Haiqiang Qi, Guifang Pan, Xuelin Shi, Zhirong Sun, Cu–Fe–FeC3@nitrogen-doped biochar microsphere catalyst derived from CuFe2O4@chitosan for the efficient removal of amoxicillin through the heterogeneous electro-Fenton process, Chemical Engineering Journal, 434 (2022) 134675.

  1. Au-s should consider to transfer a part of the illustrations on the characterization of the catalyst into the Supplementary Info.

-- If space permits, I want to keep all the catalyst characterization illustrations and descriptions, because they are all essential parts to present the materials’ catalytic properties.

Reviewer 3 Report

Review-Coatings-1670709-to Authors: This manuscript investigates the preparing course of Cu/Fe/clay/polymeric catalyst for degradation of pollutants. Starting from the Abstract the novelty, originality and the significance of the work were not clearly put forward. Also there are obvious language and typo errors. In the Introduction the argument for MB degradation is human health. That is not realistic, MB was (before) only marginally used as a typical model pollutant and the world wide quantities are hardly causing pollution. Much more relevant is the CIP pollution which is a commercially used antibiotic. Generally, I don’t see what do you want to show by putting CIP and MB in one investigation. How do they correlate. The first paragraph need to be shorter.

The use of component for catalytic purpose was only discreetly described in the into. I don’t see an explanation why would someone combine chitosan and clay, and what exactly is the expectation. Some references for adsorption of uranium were given but that is hardly relevant for MB and CIP. You did not state and advantages of the ultrasonic impregnation method.

Experimental is not acceptable because of lack of clarity. Certain amount is not a measure.

Regarding XRD results, you show the crystalline structure of the clay did not disappear, but did it change; basal peaks, interlayer distance? FTIR, at 740 and 476 I don’t see a conclusive difference between the scans, how do you explain?

For EDS, I again don’t see a conclusive difference between the scans, how do you explain?

BET was done for products not exposed to reaction. I see that as a major problem. Basically you don’t discuss what happens with clay in terms of cation exchange, swelling, mechanical degradation etc. These are critical parameters to consider when dealing with fragile natural smectite type clay.

XPS is very surface sensitive. But I don’t see which surface you targeted. The “inner” surface was not investigated. You don’t have films or spherical particles with well-defined layers, so I must say I don’t see the point of XPS.

For the catalysis tests; I don’t really see the point of comparing tests with and without peroxide when the reference is just peroxide. Also I would want to see how clay+peroxide, and polymer+peroxide behave as a reference.

The manuscript shows low novelty and average overall quality. The discussion and conclusions are not fully supported by the results. Complete reorganization of the introduction, experimental and results & discussion is necessary before the manuscript may be acceptable. I suggest major revision.

Author Response

Responses to reviewer 3’s comments:

Thanks for your useful comments and suggestions on the language and the structure of our manuscript. We have modified the manuscript accordingly, and the detailed corrections are listed below point by point:

  1. This manuscript investigates the preparing course of Cu/Fe/clay/polymeric catalyst for degradation of pollutants. Starting from the Abstract the novelty, originality and the significance of the work were not clearly put forward. Also there are obvious language and typo errors. In the Introduction the argument for MB degradation is human health. That is not realistic, MB was (before) only marginally used as a typical model pollutant and the world wide quantities are hardly causing pollution. Much more relevant is the CIP pollution which is a commercially used antibiotic. Generally, I don’t see what do you want to show by putting CIP and MB in one investigation. How do they correlate. The first paragraph need to be shorter.

-- We have rewritten the abstract and first paragraph of introduction. We gave a joint of investigation of a dye and a drug in our manuscript, which can be seen in page 2.

  1. The use of component for catalytic purpose was only discreetly described in the into. I don’t see an explanation why would someone combine chitosan and clay, and what exactly is the expectation. Some references for adsorption of uranium were given but that is hardly relevant for MB and CIP. You did not state and advantages of the ultrasonic impregnation method.

-- We have given an explanation in the manuscript why researchers combined chitosan and attapulgite and what researches they wanted to do. About the chitosan/attapulgite composite, we didn’t find any references relevant to MB and CIP degradation, only found those relevant to uranium adsorption. The advantages of the ultrasonic impregnation method are fast and effective.

  1. Experimental is not acceptable because of lack of clarity. Certain amount is not a measure.

-- We have changed those statements which are lack of clarity.

  1. Regarding XRD results, you show the crystalline structure of the clay did not disappear, but did it change; basal peaks, interlayer distance? FTIR, at 740 and 476 I don’t see a conclusive difference between the scans, how do you explain?

--We have analyzed the change of basal peaks. About FTIR, the peaks of Fe-O and Cu-O may be not too obvious due to low loading percentage, but they were really difference between the scans, please see the following figure. And FTIR test is just an assistant method for existence of Fe2O3 and CuO, we have tested Mapping, EDS and XPS of the samples to prove the existence of Fe2O3 and CuO.

  1. For EDS, I again don’t see a conclusive difference between the scans, how do you explain?

--There were obvious difference between the scans, please see following figures:

From the Figures above, we can clearly see Fe element in Figure 2 (h) and Fe & Cu elements in Figure 2 (i). The element-Ka (such as Fe-Ka) means a background for corresponding peak.

  1. BET was done for products not exposed to reaction. I see that as a major problem. Basically you don’t discuss what happens with clay in terms of cation exchange, swelling, mechanical degradation etc. These are critical parameters to consider when dealing with fragile natural smectite type clay.

--According to the references, BET tests were done for catalysts never exposed to reaction. After introducing chitosan into attapulgite, dispersibility and mechanical strength of attapulgite were improved, so the swelling of attapulgite could be ignored. When test BET, the samples are not exposed to solution, so cation exchange is not considered. If the samples are in solution, cation exchange is conducive to adsorption and catalysis processes.

  1. XPS is very surface sensitive. But I don’t see which surface you targeted. The “inner” surface was not investigated. You don’t have films or spherical particles with well-defined layers, so I must say I don’t see the point of XPS.

--We surely investigated the inner surface of the samples according to the procedure of XPS test (the samples were spherical particles and they were smashed before test).

  1. For the catalysis tests; I don’t really see the point of comparing tests with and without peroxide when the reference is just peroxide. Also I would want to see how clay+peroxide, and polymer+peroxide behave as a reference.

--Clay (attapulgite) and polymer (chitosan) were combined just for a composite support for active components Fe2O3 and CuO, that means Fe2O3 and CuO were really the effective substances to catalyze hydrogen peroxide to ·OH while clay (attapulgite) and polymer (chitosan) are not. So it didn’t show how clay+peroxide and polymer+peroxide behave.

Round 2

Reviewer 1 Report

The manuscript has been improved and addressed most of my concerns.

This work can be published in its present form.

Author Response

Thanks for your useful comments and suggestions on the language and the structure of our manuscript.

Reviewer 3 Report

Review-Coatings-1670709-R1 to Authors: This is the revised manuscript that investigates the preparing course of Cu/Fe/clay/polymeric catalyst for degradation of pollutants. Some changes in the Introduction are acceptable.

The lack of clarity and explanations in XRD is not acceptable. You must see the bigger picture from the analysis results and not just neglect dana other than basal peaks.

FTIR and EDS really just barely show some differences. In EDS figure 3 I really don’t see peaks in the highlighted area.

Regarding BET, I am not saying the BET analysis could harm the samples, but I am saying that the photocatalysis in a solution can. Can you show that exposing samples to CIP and MB solution will not cause reactions?

For XPS the crushing is meaningful only slightly more than without crushing, but that is hardly sufficient.

Regarding catalytic tests, the Fe2O3 and CuO are broadly reported, but the composite support is not. So I do think it is important to say as much as possible for the support also, despite there are no active materials in the support. Si I do think the support reference is necessary.

The manuscript is somewhat upgraded, but still far from the acceptable version. I suggest major revision.
